# LeRobot: An Open-Source Library for End-to-End Robot Learning

**Remi Cadene**[*]
Hugging Face

**Simon Aliberts**[*]
Hugging Face

**Francesco Capuano**[* †]
University of Oxford

**Michel Aractingi**[*]
Hugging Face

**Adil Zouitine**[*]
Hugging Face

**Pepijn Kooijmans**[*]
Hugging Face

**Jade Choghari**[*]
Hugging Face

**Martino Russi**[*]
Hugging Face

**Caroline Pascal**[*]
Hugging Face

**Steven Palma**[*]
Hugging Face

**Mustafa Shukor**[*]
Hugging Face

**Jess Moss**[*]
Hugging Face

**Alexander Soare**[*]
Hugging Face

**Dana Aubakirova**[*]
Hugging Face

**Quentin Lhoest**
Hugging Face

**Quentin Gallouédec**
Hugging Face

**Thomas Wolf**
Hugging Face

## Abstract

Robotics is undergoing a significant transformation powered by advances in high-level control techniques based on machine learning, giving rise to the field of robot learning. Recent progress in robot learning has been accelerated by the increasing availability of affordable teleoperation systems, large-scale openly available datasets, and scalable learning-based methods. However, development in the field of robot learning is often slowed by fragmented, closed-source tools designed to only address specific sub-components within the robotics stack. In this paper, we present `lerobot`, an open-source library that integrates across the entire robot learning stack, from low-level middleware communication for motor controls to large-scale dataset collection, storage and streaming. The library is designed with a strong focus on real-world robotics, supporting accessible hardware platforms while remaining extensible to new embodiments. It also supports efficient implementations for various state-of-the-art robot learning algorithms from multiple prominent paradigms, as well as a generalized asynchronous inference stack. Unlike traditional pipelines which heavily rely on hand-crafted techniques, `lerobot` emphasizes scalable learning approaches that improve directly with more data and compute. Designed for accessibility, scalability, and openness, `lerobot` lowers the barrier to entry for researchers and practitioners to robotics while providing a platform for reproducible, state-of-the-art robot learning.

## 1 Introduction

Early successes in robotics relied on the precise description of robot-environment interactions, typically consisting in analytical descriptions of rigid-body kinematics, contact modeling, and planning under uncertainty (*explicit models*). While effective in controlled settings, deriving accurate models for diverse deployment scenarios is difficult and error-prone, often requiring substantial expert effort and thus has limited scalability. Recent advances in Machine Learning (ML) have catalyzed a shift toward tackling robotics problems with *implicit models*, typically *learned* rather than formulated.

A key advantage of learning-based methods (*implicit models*) is their scalability: performance empirically improves with larger datasets, and more compute. In turn, the shift from explicit to implicit models promises to address many of the challenges holding robotics back: rather than hand-tuning the different components of a typical robotics pipeline, robot learning algorithms learn monolithic

---

[*]Core team. [†] Work done while at Hugging Face.

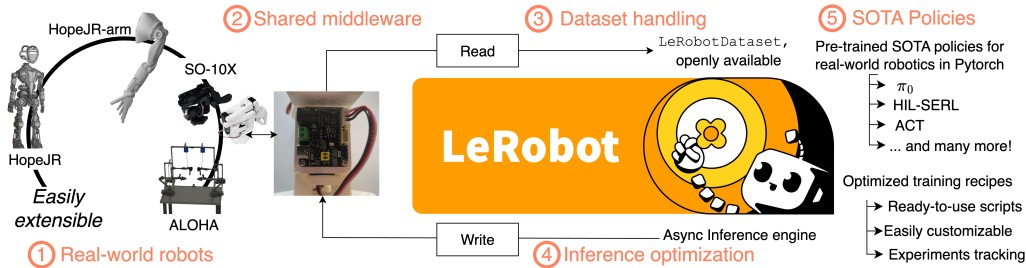

Figure 1: `lerobot` is an open-source library for end-to-end robot learning. It covers the entire stack, from middleware motor interfaces to large-scale data collection and dataset streaming, supporting an optimized inference stack, scalable implementations of SOTA robot learning algorithms, and providing support for training custom models as well as easily reusing pre-trained ones.

control policies end-to-end, adapting to different input modalities and typically improve with increasing quantities of data, echoing broader trends in vision, language, and multimodal learning.

Despite this momentum, the robot learning ecosystem is fragmented as (1) high-to-low level control interfaces (*middleware*) are often tailored to specific robots and difficult to adapt, and (2) datasets lack common formats and tooling, resulting in robot and task-specific contributions that are difficult to reproduce and use in practice. `lerobot` is an open-source library providing a unified, end-to-end stack for robot learning, and it is vertically integrated across the stack featuring:

- **Unified robot integration.** A consistent, Python-based middleware API for real-world motor control across diverse platforms, bridging typical ML frameworks and real-world robotics across a variety of robots, ranging from low-end manipulators to humanoid arms and hands.

- **Standardized datasets.** An efficient, multimodal format for recording, storing, and streaming high frame-rate sensory and image data via `LeRobotDataset`, a custom dataset format built for scale. With seamless integration into the open-source ecosystem, `LeRobotDataset` encourages openness and research reproducibility.

- **Optimized inference.** An optimized inference stack that decouples action planning from control execution both (1) physically and (2) logically, enabling policies to (1) run on separate machines with increased computational resources compared to those onboard robots, and (2) in parallel with low-level control loops, for robust deployment and dynamic adaptability at runtime.

- **Efficient, reusable algorithms.** Clean, PyTorch-based implementations of state-of-the-art (SOTA) robot learning methods, optimized for (1) training custom models from scratch and (2) using openly-available pre-trained models.

Together, these components address fragmentation issues in the field, reducing the barrier to entry for robotics by providing vertical integration across the entire robot learning stack, with a clear emphasis on accessibility and scalability, aiming at accelerating progress in the field.

## 2 BACKGROUND

### 2.1 EXPLICIT AND IMPLICIT MODELS

Autonomous motion leverages either *explicit* or *implicit* models (Bekris et al., 2024). Classical robotics historically uses *explicit models*, implemented as modular pipelines for perception, planning, and control (Siciliano & Khatib, 2016). This approach suffers from compounding errors, poor scalability to diverse deployment scenarios, and *undermodeling issues* due to simplified analytical models of physical interactions, limiting its effectiveness to unstructured, dynamic environments (e.g., a house versus a factory line). In contrast, robot learning relies on *implicit models* to develop monolithic, data-driven policies directly mapping observations to action. Robot learning also priori-

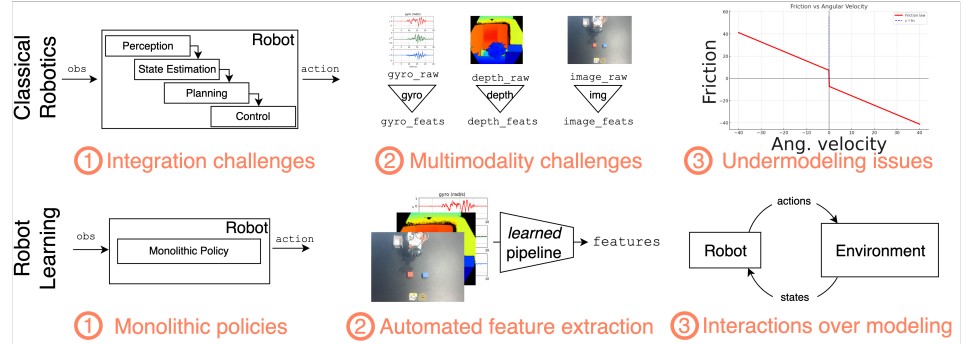

Figure 3: Classical robotics uses modular, model-based pipelines with hand-crafted features, while robot learning employs monolithic, data-driven policies that learn directly from interaction data.

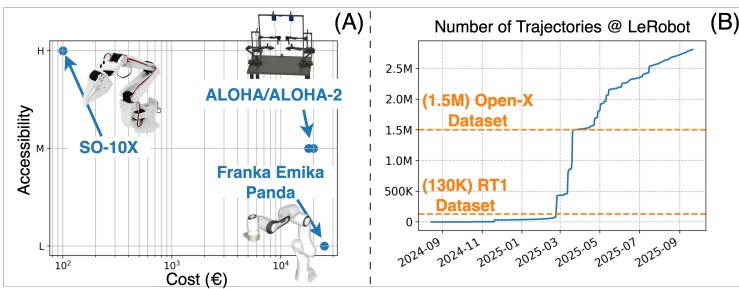

Figure 4: (A) Low-cost, open-source robots like SO-10X and ALOHA cost a fraction of proprietary industrial arms, using consumer-grade parts and 3D-printable designs. (B) Decentralized efforts to collect expert demonstrations in the form of trajectories surpassed centralized efforts for the collection of large amounts of real-world robotics data.

tizes interaction data over rigid assumptions, and replaces hand-engineered components with learned representations, offering a more robust and adaptable solution for unstructured environments.

The adaptability of these learned, implicit models stems directly from their scalability with data—a primary advantage over classical approaches. In this context, real-world robotics data is often collected in the form of expert demonstrations via *teleoperation*, a process where *"cognitive decisions are made by [a] human user, while the robot is responsible for their mechanical implementation"* (Siciliano & Khatib, 2016, Ch.43, §1). In recent years, teleoperation hardware has become increasingly affordable, making it more and more relevant for robot learning, either by teleoperating in virtual reality (VR) or in the real world. Consumer-grade VR teleoperation headsets for robotics have been used to collect robot data both on real-world and simulated robots (Bjorck et al., 2025), and low-cost teleoperated robotic arms (Zhao et al., 2023; Aldaco et al., 2024; Wu et al., 2024; Knight et al., 2024) are increasingly empowering researchers and practitioners to collect real-world robotics data. In turn, this results in a multiplication of centralized (Brohan et al., 2023; Collaboration et al., 2025; Khazatsky et al., 2025) and de-centralized (Section 3.2) efforts to collect robot data. Figure 4 shows how fully accessible teleoperated platforms such as the SO-100, SO-101 (jointly referred to as SO-10X, (Knight et al., 2024)) and ALOHA-2 (Aldaco et al., 2024) can cost down to a fraction of closed-source, industrial-grade robots such as the Franka Emika Panda arm. Consequently, these low end robot platforms can be used to collect large amounts of data in a decentralized effort powered by the very accessibility—low-cost, open designs, 3D-printable parts—of these low-end robot platforms.

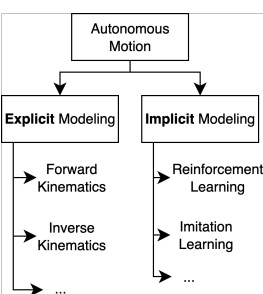

Figure 2: Some of the explicit and implicit models for autonomous motion.

## 2.2 ROBOT LEARNING

**Reinforcement Learning** Reinforcement learning (RL) (Sutton & Barto, 2018) has been extensively applied to robotics (Kober et al., 2013), for the inherently sequential nature of control problems and Deep RL's capability to learn strategies for return maximization $\max_\pi J(\pi) = \max_\pi \mathbb{E}_{\tau \sim \pi}\left[\sum_{t=0}^{T} \gamma^t r_t\right]$ directly from highly-dimensional, unstructured observations such as images (Mnih et al., 2013). Off-policy, entropy-regularized methods such as Soft Actor Critic (Haarnoja et al., 2018) can be adapted to exploit teleoperation data and safely train in the real-world, thereby sidestepping concerns related to operative safety and simulation-induced discrepancies. Reinforcement Learning with Prior Data (RLPD) (Ball et al., 2023) mixes offline and online buffers without pretraining to speed up convergence, and in conjuction with (1) learned reward classifiers overcoming the need to define brittle hand-crafted rewards (Luo et al., 2025) and (2) targeted human interventions during training, can yield near-perfect success rates in challenging manipulation tasks within 1-2 hours of real-world training (HIL-SERL, Luo et al. (2024)).

**Imitation Learning** Imitation Learning via Behavioral Cloning (BC) offers a pragmatic alternative to real-world RL by learning control directly from human demonstrations, eliminating the need for reward design and reducing exploration risk by learning to reproduce the behavior of an expert demonstrator (Pomerleau, 1988). Collected via teleoperation on increasingly affordable hardware, large corpora of robotics data also enable training at a scale across tasks and embodiments (Collaboration et al., 2025; Khazatsky et al., 2025). BC relies on learning *generative* models of the joint (or conditional) distribution over state-action pairs $p : \mathcal{S} \times \mathcal{A} \mapsto [0, 1]$, $p(a, s)$ (or $p(a|s)$) to learn from data distributions exhibiting multiple modes, such as teleoperation data (Florence et al., 2022). Recent works in BC thus employ powerful generative models to learn the conditional distribution $p(a|s)$, learning from multimodal demonstrations and produce coherent action sequences: Zhao et al. (2023) leverages (conditional) Variational Auto-Encoders (Kingma & Welling, 2022; Sohn et al., 2015), Chi et al. (2024) relies on Diffusion Models (Ho et al., 2020) whereas Black et al. (2024); Shukor et al. (2025) both rely on Flow Matching (Lipman et al., 2023). Inspired by successes in developing foundation models for vision (Dosovitskiy et al., 2020) and language (OpenAI, 2024), BC is also increasingly being used in efforts aiming to develop *robot foundation models* (Jang et al., 2022; Brohan et al., 2023; Black et al., 2024), scaling up both data and compute used to learn visuomotor policies suitable for real-world deployment across tasks and even robot embodiments.

Robot learning algorithms are often implemented as standalone components and their integration with the rest of the robotics stack remains challenging.

## 2.3 PRACTICAL CHALLENGES FOR ROBOT LEARNING RESEARCH

Despite scientific advances, the robot learning ecosystem remains fragmented, impeding reproducibility and raising the barrier to entry for research.

- **Disaggregated Middleware:** While middleware abstractions are available, it is common to encounter middleware components tailored to specific platforms in practice. This heterogeneity often forces teams to develop bespoke adaptations, siloing efforts.

- **Datasets and Formats:** Large-scale datasets are typically shared in a different formats. Data is often released in varied formats like TensorFlow Datasets, ROS bags, or bespoke JSON layouts. The absence of a universal, modality-rich schema prevents the seamless aggregation of disparate datasets into larger mixtures.

- **Learning Frameworks:** The deep learning literature has consistently demonstrated that minor implementation differences in algorithms, data handling, and evaluation pipelines can lead to significant variance in results (Henderson et al., 2018). In robotics, these issues are compounded by hardware variability, further hindering reproducibility.

This ecosystem-wide fragmentation imposes significant incidental complexity on researchers, diverting resources from core scientific inquiry to systems integration. `lerobot` addresses these limitations by providing an end-to-end, open, and scalable library designed to unify hardware interfacing, collecting and streaming data, and training and deploying advanced policies with minimal engineering overhead.

| | Robot Type | Cost (€) |
|---|---|---|
| SO-100/101 | Manipulator (Bimanual) | ∼ 225 (550) |
| Koch-v1.1 | Manipulator (Bimanual) | ∼ 670 (1346) |
| ALOHA | Bimanual Manipulator | ∼ 21k |
| HopeJR-Arm | Humanoid Arm and hand | ∼ 500 |
| LeKiwi | Mobile Manipulator | ∼ 230 |

(a) Cost for all robot platforms supported by `lerobot` and with an openly-available Bill Of Materials (BOM).

| Robot | # Downloads | # Datasets | # Episodes |
|---|---|---|---|
| Panda | 1'878'395 | 588 | 926'776 |
| xArm | 1'107'329 | 74 | 450'329 |
| WidowX | 832'177 | 100 | 214'117 |
| KUKA | 662'550 | 3 | 419784 |
| SO-101 | 319'586 | 3'965 | 58'299 |
| SO-100 | 278'697 | 5'161 | 78'510 |
| Koch-v1.1 | 43'561 | 849 | 20'959 |

(b) All Top-4 robots for number of downloads, datasets and episodes openly shared, listed in decreasing order by the total number of downloads.

## 3 FEATURES

`lerobot` is designed for accessibility, scalability, and reproducibility in robot learning. The library natively integrates (1) entirely open-source hardware platforms costing a fraction of closed-source devices, (2) a unified middleware shared across low-level robot interfaces (3) data collection, storage and streaming tools, (4) an optimized inference engine and (5) many ready-to-use implementations of SOTA methods in robot learning, useful to both train models from scratch and re-use openly available pre-trained models. `lerobot` is entirely open-source, and highly accessible due to its reliance on low-cost teleoperation kits, focus on empowering large-scale datasets via streaming, and simple interface to adopt models in fully reproducible pipelines.

### 3.1 ACCESSIBLE REAL-WORLD ROBOTS

`lerobot` currently supports multiple real-world robot platforms for both static and mobile manipulation. The library fully integrates the SO-100 and SO-101 arms (Knight et al., 2024), both in a single and bimanual setup. The library also supports the Koch-v1.1 (Moss, 2025) and ALOHA-2 (Aldaco et al., 2024) manipulators, the Hope-JR humanoid arm (TheRobotStudio, 2025), the Stretch-3 (Hello Robot, 2025) and LeKiwi (SIGRobotics-UIUC, 2025) mobile manipulation platforms, and lastly the Reachy-2 humanoid (Mick et al., 2019). `lerobot` is designed to interface multiple open devices with a shared middleware that can be used to (1) read the configuration on a *leader* robot and write it on *follower* robot for teleoperation and (2) directly control the *follower* with a learned policy.

Table 1a shows the cost for all the robot platforms currently supported by `lerobot` with an openly-available Bill of Materials (BOM), reported for completeness in Appendix A. `lerobot` can support multiple robot platforms thanks to a shared middleware embedded in higher-level abstractions for the different robots supported, and engineered to interface directly with the low-level SDKs of major low-cost actuator producers (FeeTech and Dynamixel). Crucially, the middleware is designed to be easily extensible and highly composable. We refer to Appendix B for an example of teleoperation using `lerobot`.

### 3.2 DATASETS

To address the fragmented nature of data in robotics research, we introduce `LeRobotDataset`, `lerobot`'s unified multimodal dataset schema. This standardized format is engineered to provide convenient and standardized access to robotics data spanning diverse modalities, including high-frequency sensorimotor readings, multiple camera feeds, and teleoperation status signals. The schema is designed to be self-contained, accommodating general metadata such as textual descriptions of the demonstrated tasks for filtering and language-conditioned policies, specifics of the robot embodiment considered, and relevant experimental parameters such as frames-per-second (FPS) of data capture and the types sensors used. As of September 2025, 16K+ datasets from 2.2K+ individual contributors are openly shared via the `LeRobotDataset` format, featuring robots directly integrated in the library such as the SO-10X arm and unsupported robots (Franka Emika Panda, xArm, R1Pro), ported to the `LeRobotDataset` format by the open-source community. We argue the support of different robot configuration underscores the flexibility of our dataset format, and that the coexistence of both large-scale academic benchmarks and small-scale data collection efforts exemplifies the breadth of use-cases that our dataset format can accomodate.

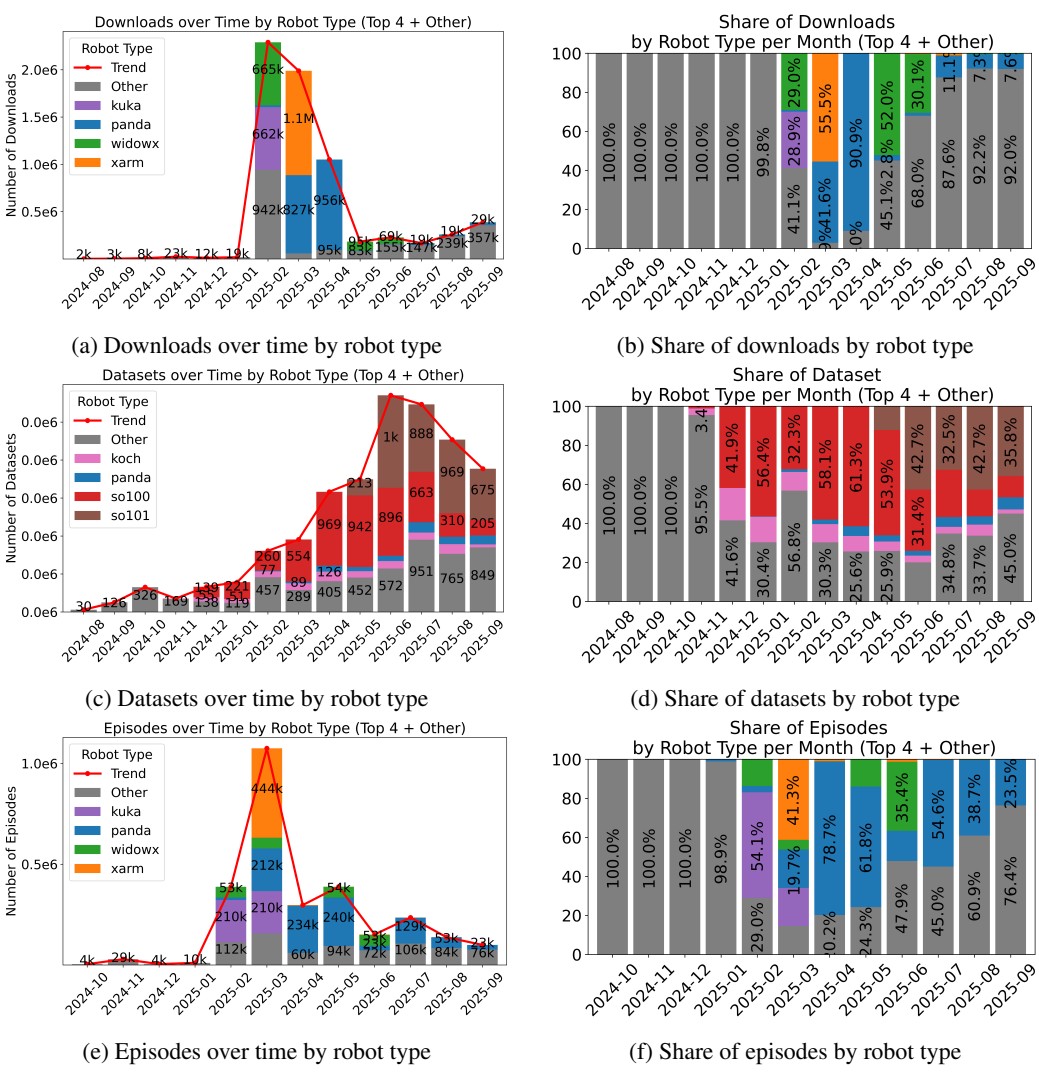

Figure 5: Numbers and trends of downloads, datasets, and episodes by robot type over time. The number of episodes in each dataset has been explicitly tracked starting in October 2024 only. For completeness, we report the top-5 robots grouped in *Other*, for each of the metrics considered, in Table 4.

Open datasets are available for downloads, and Figure 5a shows the evolution of the number of downloads over time, with a breakdown of the share of downloads per robot type (Table 1b) and per robot type over time (Figure 5b, see Appendix C for further details). Despite `lerobot` only supporting a limited number of robots (grouped under the *Other* tag in Figure 5a and Figure 5e), datasets collected for other platforms such as the Franka Emika Panda and xArm lead in the number of downloads and size of the datasets collected (Figure 5e). We argue this follows from these platforms being often featured in research-oriented centralized data collection efforts (Collaboration et al., 2025; Khazatsky et al., 2025). Conversely, platforms such as the SO-10X are increasingly featured in small-scale decentralized community efforts powered by the accessibility of (1) the hardware platforms used and (2) `LeRobotDataset` format, with 50%+ of the datasets contributed being collected directly on the SO-10X platforms (Figure 5d).

A primary design principle of the dataset format is scalability. The dataset architecture is optimized to handle large-scale repositories potentially containing millions of expert trajectories. This unified interface for multi-modal, sequential data is designed for seamless integration with the Py-Torch ecosystem, further promoting standardized and repeatable research workflows. This design

|  | # Params | Peak Memory | | | |
|---|---|---|---|---|---|
|  |  | CPU | MPS | RTX 4090 | A100 |
| ACT | 52M | 817.4MB | 462MB | 211.24 MB | 211.24 MB |
| Diffusion Policy | 263M | 1.22GB | 224MB | 1.12 GB | 1.12 GB |
| $\pi_0$ | 3.5B | 4.13GB | 97MB | 13.32 GB | 13.32 GB |
| SmolVLA | 450M | 1.69GB | 555MB | 1.75 GB | 1.75 GB |

Table 2: Peak memory consumption of policy models currently supported by `lerobot`. All models are run in full precision (fp32). Diffusion and Flow Models are run with 10 denoising steps at inference. All models maintain their original outputs shapes.

is complemented by a native streaming capability designed to enhance accessibility: users can process remotely-hosted large-scale datasets without the prerequisite of downloading the entire corpus, thereby lowering barriers to entry for the broader community and improving on the accessibility of robot learning research. See Appendix C for more details on streaming.

```python
from lerobot.datasets.lerobot_dataset import LeRobotDataset
from lerobot.datasets.streaming_dataset import StreamingLeRobotDataset

repo_id = "lerobot/svla_so101_pickplace"
# Downloads the whole dataset and loads it in memory
# (allowing for random access)
dataset = LeRobotDataset(repo_id)

# Streams frames on the fly without downloading
# (access frames sequentially, .next())
dataset = StreamingLeRobotDataset(repo_id)
```

## 3.3 MODELS

`lerobot` supports reference implementation for multiple SOTA robot learning algorithms, providing useful baselines for experimentation and accessible models across RL, such as HIL-SERL (Luo et al., 2024) and TD-MPC (Hansen et al., 2022) and BC, both for single-task ACT (Zhao et al., 2023), Diffusion Policy (Chi et al., 2024) and VQ-BET (Lee et al., 2024), and multi-task models such as $\pi_0$ (Black et al., 2024) and SmolVLA (Shukor et al., 2025) (Figure 6).

`lerobot` offers support for custom models too, grouped together under the *Other* tag in Figure 7. All the control policies implemented in `lerobot` are written in pure Pytorch (Paszke et al., 2019), and integrated with the library to allow (1) training models from scratch on datasets collected via real-world demonstrations, and (2) inference using openly available pre-trained models. The library is designed to for high accessibility, providing a composable set of recipes which can be used to train a model from scratch in less than 100 lines-of-code (LOC), and serve models in less than 40 LOC (Appendix D).

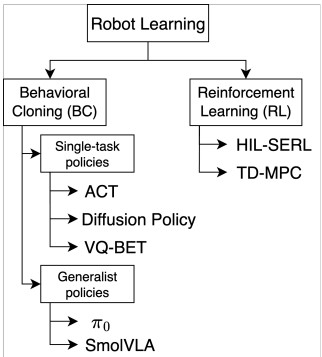

Figure 6: The different robot learning algorithms currently supported by `lerobot`.

In its effort to foster accessibility, `lerobot` supports multiple models with different computational constraints, ranging from lightweight single-task models to larger, multi-task models. ACT (Zhao et al., 2023) is a particularly popular model dominating the number of uploads (Figure 7a), consistenly ranking as one of the most popular policies trained (Figure 7b) and used (Figure 7d). We ascribe the popularity of ACT to (1) its small size and fast inference speed and (2) straightforward application to limited amount of real-world demonstrations, allowing users to obtain well-performing policies with as little as 50 real-world trajectories. As a single-task model, however, ACT necessitates retraining whenever changes in the experimental conditions occur. SmolVLA (Shukor et al., 2025) is a powerful, small-scale Vision-Language-Action model which allows to control real-world robots via language conditioning, resulting in an overall wider applicability to practical scenarios.

| | | Avg Inference Latency (ms) | | | |
| --- | --- | --- | --- | --- | --- |
| | **# Params** | **CPU** | **MPS** | **RTX 4090** | **A100** |
| ACT | 52M | 182.313 ± 40.82 | 42.667 ± 10.085 | 5.013 ± 0.061 | 13.77 ± 0.445 |
| Diffusion Policy | 263M | (100%) | 3453.838 ± 39.271 | 369.788 ± 0.193 | 613.893 ± 10.173 |
| $\pi_0$ | 3.5B | (100%) | (100%) | 209.381 ± 2.762 | 568.978 ± 2.937 |
| SmolVLA | 450M | 2028.461 ± 302.59 (2%) | 721.826 ± 57.748 | 99.244 ± 1.195 | 278.833 ± 1.886 |

Table 3: Average and standard deviation inference latency over 100 forward passes for policy models currently supported by lerobot. Diffusion and Flow Models are run with 10 denoising steps at inference time. (x%) indicates the percentage of samples that timed-out before the 5000ms hard stop (0% omitted).

Table 2 and Table 3 report the peak memory footprint and the average inference latency, measured over 100 test samples, for the most widely used policies supported by `lerobot`. Evaluations were conducted on four platforms: (1) a MacBook Pro M1 (2021, 16GB, CPU only), (2) the same MacBook Pro with the MPS backend, (3) an NVIDIA RTX 4090, and (4) an NVIDIA A100. All models were executed in full `fp32` precision at runtime, with inference timed-out after 5 seconds. Overall, peak memory footprints largely align with theoretical estimates obtained from the combination of model parameter count and numerical precision. The main exceptions are the CPU and MPS backends, where unified memory and frequent offloading to swap introduce variability, obscuring direct performance comparisons and increasing latency. Latency measurements are averaged across all non—timed-out trials, with both mean and standard deviation reported in Table 3. Smaller, task-specific models such as ACT exhibit high efficiency on accelerated backends like MPS and achieve inference rates of ∼100-200Hz on high-end GPUs such as the RTX 4090 and A100. Crucially, larger models such as $\pi_0$ require substantially longer per each forward passes on average on all platforms, and even fail to complete inference within the 5s limit on lower-tier devices, underscoring the challenges in deploying robotics foundation models in practice.

## 3.4 INFERENCE

`lerobot` defines a custom inference stack which is designed to decouple action prediction (*inference*) from action execution (*control*), at both the physical and logical level (Figure 8). This optimized stack is designed for modern robot learning policies, increasingly predicting sequences of actions (*action chunks*, $a_{t:t+H-1}$, (Zhao et al., 2023)) rather than single controls. All the BC policies supported by `lerobot` predict action chunks.

*Physical* decoupling allows inference to run on a remote machine connected over the network to the robot's low-level controller. This design enables the use of higher-end computational resources than those typically available aboard a robot for inference, while control is maintained at the desired control frequency stepping through the multiple actions received. Further, *logical* decoupling implements inference via an *asynchronous* producer-consumer scheme: the inference process predicts *action sequences* with a look-ahead horizon $H$ *in parallel* with environment control, which consumes actions at a fixed control rate. Overlapping predictions are merged via a generalized aggregation function $f$, which users can easily specify for their own use cases, ensuring a non-empty action queue and preventing idleness of the robot by overlaying action prediction and action execution. We refer to Appendix E for more details on the performance of decoupled inference.

## 4 SIMULATION

While the core focus of `lerobot` is to lower the barrier to entry to enable real-world robotics applications, `lerobot` does also support different simulation environments for benchmarking purposes. In practice, simulation proves challenging for the kind of contact-rich, complex tasks `lerobot` targets. This justifies the library's choice to train as much as possible on real-world data, relying on simulation *primarily for the systematic evaluation* of robot learning algorithms. To enable this, we

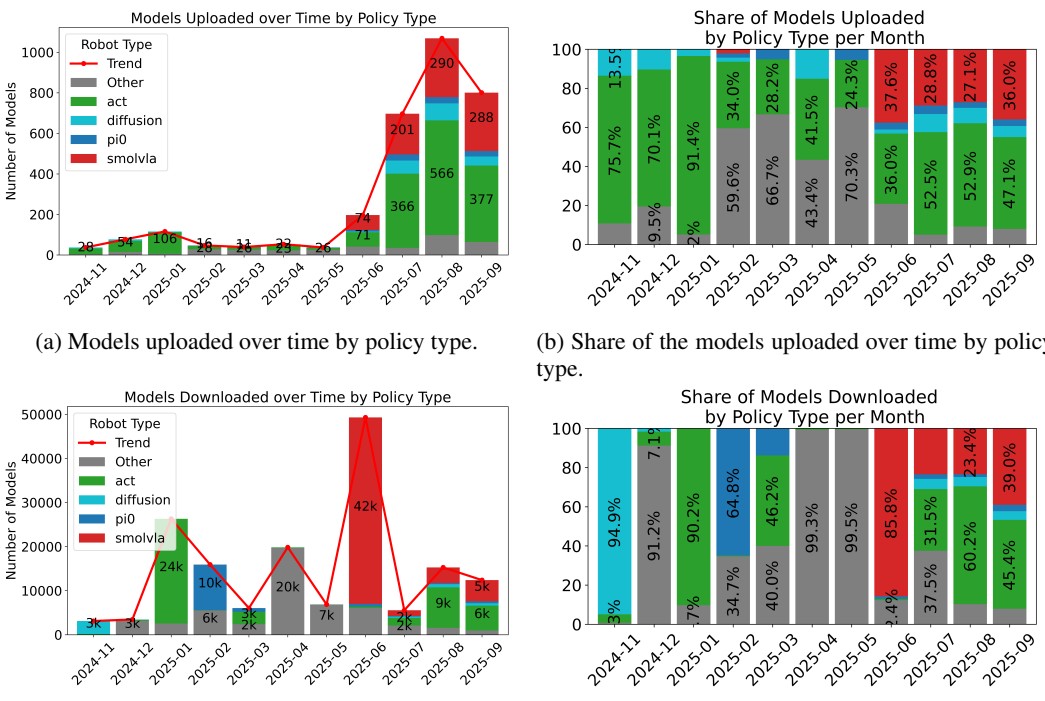

(a) Models uploaded over time by policy type.

(b) Share of the models uploaded over time by policy type.

(c) Models downloaded over time by policy type.

(d) Share of models downloaded over time by policy type.

Figure 7: Numbers and trends of uploads and downloads of robot learning models by policy type over time. TD-MPC (Hansen et al., 2022), HIL-SERL (Luo et al., 2024) and VQ-BET (Lee et al., 2024) are absent from all visualizations as they are not typically uploaded by users.

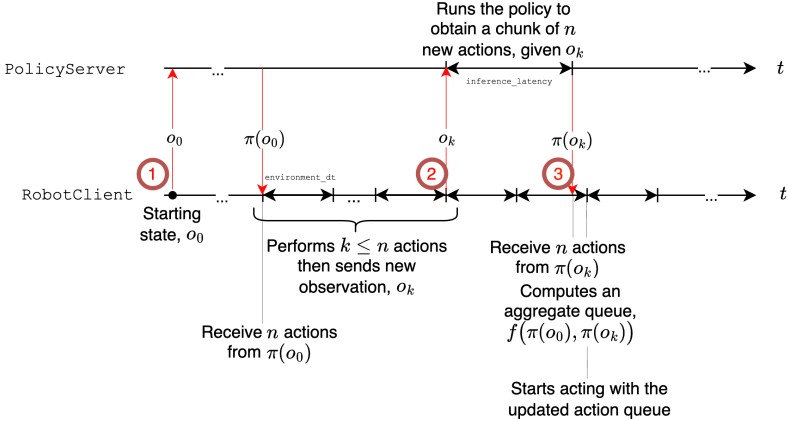

Figure 8: Overview of the generalized inference schema supported by `lerobot`, whereby a remote server can be used to host compute-expensive policies for inference, while the robot client receives a stream of the actions chunks to enact. The schema provides scalability and flexibility through the possibility to fully customize the function $f$ used to aggregate overlapping chunks.

provide evaluation support via the `lerobot` API for both LIBERO (Liu et al., 2023) and Meta-World (Yu et al., 2020), two popular simulation environments that are often used as benchmarks for robot learning research (Black et al., 2024; Shukor et al., 2025).

**LIBERO** While developed to specifically assess the life-long learning capabilities of generic autonomous agents, LIBERO is also used in robot learning research as a benchmark to demonstrate

the adaptability and performance of novel methods, e.g. in manipulation settings (Black et al., 2024; Shukor et al., 2025). While accomodating for procedural task generation, LIBERO proposes four fixed task suites with 10 tasks each researchers can benchmark on. Task suits are developed to quantify the amount of information that is shared under different conditions in terms of spatial arrangement (LIBERO-SPATIAL), object considerd (LIBERO-OBJECT), and overall task variations (LIBERO-GOAL). LIBERO does also provide a benchmark for continuing, short (LIBERO-90) and long-horizon (LIBERO-LONG) tasks requiring the transfer of entangled knowledge between the aforementioned sources of variation. Typical LIBERO evaluation protocols report the success rate over a number of test episodes, and `lerobot` natively integrates LIBERO.

**Meta-World**  Similarly to LIBERO, Meta-World was first developed as a benchmark for assessing the performance of generic autonomous systems, with a particular focus on fast adaptation to novel scenarios via meta-learning. Adapting quickly to novel scenarios is a particularly promising area of research in the field of robotics, as it holds the premise of enabling the development of systems that can effectively generalize to unseen tasks leveraging previously acquired information. The Meta-World benchmark consists of 50 distinct robotic manipulation tasks that can be combined into different benchmark suites. The benchmark is structured to quantify performance under different learning regimes: multi-task learning (MT10, MT50) where the agent learns multiple tasks simultaneously with access to a task identifier, and meta-learning (ML1, ML10, ML45) which assesses the agent's ability to adapt to new tasks using minimal data. All 50 different tasks require the same robotic arm in the same setup to interact with multiple objects with different shapes and diverse uses. Critically, all the high-level tasks presented in Meta-World require the robot to execute a combination of fixed, more fundamental skills such as reaching for an object or manipulating it. Such a common task conceptual structure proves instrumental in providing a shared interface for autonomous agents to use and learn how to transfer knowledge across different tasks: a key property of the adaptability that is required of modern robot policies (Black et al., 2024; Shukor et al., 2025).

## 5    CONCLUSIONS

In this work we introduced `lerobot`, a unified, open-source stack for end-to-end robot learning that bridges low-level control, large-scale data tooling, and scalable learning algorithms. We showed how accessible teleoperation of multiple real-world robot through a shared middleware can be used to collect real-world data across a variety of robot platforms. Further, we illustrated how standardized datasets can be exploited to collect and reuse data at scale, powering advancements in robot learning thanks to the thousands of datasets collected, resulting in hundreds of thousands of episodic data, and hundreds of models openly contributed by the robot learning community.

**Limitations**  We identify several limitations remaining in our contribution. First, robots coverage is currently far from exhaustive, as we support a practical but incomplete set of arms, grippers, sensors, and controllers. Over the course of 2025, `lerobot` went from supporting 3 manipulation setups (Koch-v1.1, SO-100, ALOHA) to the 8 regular, humanoid and mobile manipulators currently supported, and we highlight that keeping a similar rate of progress is paramount to properly serve the robot learning community. Second, the coverage in terms of robot learning algorithms is also non-exhaustive. We provide strong, reproducible implementations across key paradigms, while extending the library with additional algorithms remains future work. Third, achieving strong practical inference performance still requires low-level optimization (quantization, graph compilation, etc) that are currently disregarded by the library. We view these limitations as concrete, tractable avenues for community contributions and future development, and in the very spirit of open-source, invite the broader robot learning community to address them. However, despite these limitations, our work takes a significant step toward an end-to-end stack for robot learning, providing a useful tool for researchers and practioners in the field.

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

## A  OPENLY-AVAILABLE ROBOTS

- **SO-10X** Guide from Knight et al. (2024): `https://github.com/TheRobotStudio/SO-ARM100`.
- **Koch-v1.1** Guide from Moss (2025): `https://github.com/jess-moss/koch-v1-1`
- **ALOHA** Guide from Zhao et al. (2023) here.
- **HopeJR-Arm** Guide from TheRobotStudio (2025): `https://github.com/TheRobotStudio/HOPEJr/blob/main/Arm/BOM.md`
- **LeKiwi** Guide from SIGRobotics-UIUC (2025): `https://github.com/SIGRobotics-UIUC/LeKiwi/blob/main/BOM.md`

## B  REAL-WORLD ROBOTS API

```python
from lerobot.teleoperators.so100_leader.so100_leader import \
    SO100Leader
from lerobot.teleoperators.so100_follower.so100_follower import \
    SO100Follower

teleop = SO100Leader()
# (provided teleop matches) can also be Reachy-2, LeKiwi, etc.
robot = SO100Follower()

teleop.connect()
robot.connect()

action = teleop.get_action()
```

| Robot | # Datasets |
|---|---|
| unknown | 2370 |
| lekiwi | 535 |
| arx5 | 371 |
| aloha | 334 |
| aiworker | 202 |

| Robot | # Downloads |
|---|---|
| unknown | 711729 |
| google_robot | 438560 |
| so101 | 319586 |
| so100 | 278697 |
| aloha | 45219 |

| Robot | # Episodes |
|---|---|
| google_robot | 213852 |
| unknown | 170706 |
| so100 | 78510 |
| so101 | 58299 |
| easo | 45652 |

(a) Top-5 robot platforms in the "Other" category for number of datasets.

(b) Top-5 robot platforms in the "Other" category for number of downloads.

(c) Top-5 robot platforms in the "Other" category for number of episodes.

Table 4: Breakdown of the *Other* category by top-5 robot platforms across datasets, downloads, and episodes.

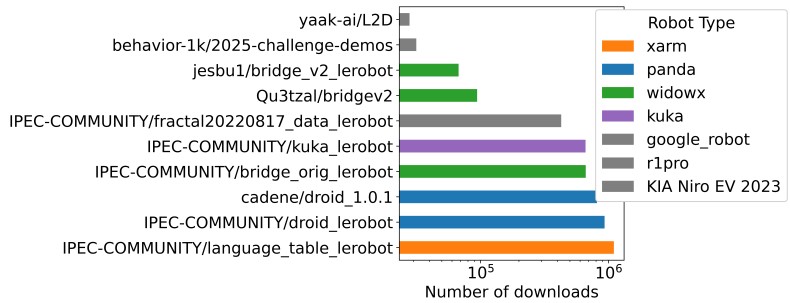

Figure 9: Openly-available datasets with the largest number of downloads using the `LeRobotDataset` format. The most downloaded datasets are academic benchmarks released by the research community (Collaboration et al., 2025; Khazatsky et al., 2025).

```python
print(action)
# {'shoulder_pan.pos': 84.74,
#  'shoulder_lift.pos': 4.95,
#  'elbow_flex.pos': 70.6,
#  'wrist_flex.pos': -88.41,
#  'wrist_roll.pos': 57.89,
#  'gripper.pos': 4.13}

robot.send_action(action)   # moves robot according to `action`
```

## C  DATASETS

Table 4 further breaks down the *Other* category for the number of downloads, datasets and episodes, and it shows how faulty dataset that do not explicitly record the robot platform used (tagged as *unknown*) dominate in the *Other* category.

Figure 9 shows the most downloaded datasets by robot type. Crucially, the largest number of downloads is not achieved for a platform natively integrated in `lerobot`, further undescoring the adoption of the `LeRobotDataset` format in the robotics community.

### C.1  STREAMING DATASETS

The development of `StreamingLeRobotDataset` addresses several fundamental challenges associated with the efficient utilization of large-scale robotic datasets in robot learning pipelines. Traditional approaches to dataset handling rely on pre-loading data into local memory, which becomes increasingly impractical as datasets grow to the million-episodes scale. `StreamingLeRobotDataset` supports a streaming paradigm, whereby *frames*—defined as individual items in a dataset—are fetched on-demand from remote storage rather than preloaded in their entirety. This architectural shift required addressing three core challenges: (1) efficient data access under strict memory constraints, (2) ensuring sufficient randomness during iteration to sup-

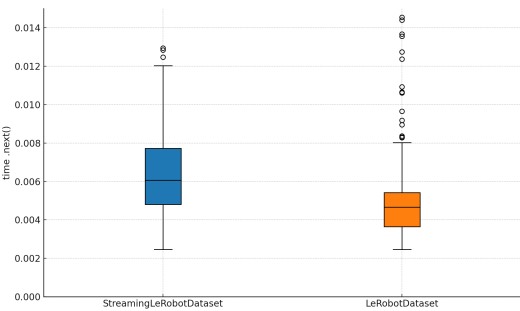 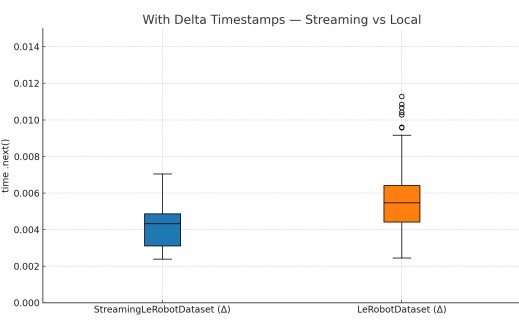

(a) Timing performance of stepping through single frames of a `StreamingLeRobotDataset` compared to a pre-loaded `LeRobotDataset`.

(b) Timing performance of stepping through a dataset retrieving multiple frames of a `StreamingLeRobotDataset` compared to a pre-loaded `LeRobotDataset`.

Figure 10: Timing performance of `StreamingLeRobotDataset` versus a regular `LeRobotDataset`.

port robust learning, and (3) enabling multi-frame retrieval in a setting that is inherently sequential and non-indexable.

**Efficient Streaming of Large-Scale Data.** The `LeRobotDataset` format partitions robotic data into tabular records (`.parquet` files) and compressed videos (`.mp4` files), alongside lightweight metadata. Metadata files are downloaded fully due to their negligible size relative to the dataset, but all high-volume video and control streams are processed on demand. This is achieved through two principal design choices: (1) adoption of an `IterableDataset` interface, and (2) integration with `torchcodec` for on-the-fly video decoding. These components together enable data consumption through simple iterative calls, while maintaining memory usage bounded irrespective of dataset size. Provided a good network connectivity, Figure 10 shows timing performance is comparable between the two formats in the steady-state regime (after initialization).

### C.2 EXAMPLE: USE A DATASET

```python
import torch
from lerobot.datasets.lerobot_dataset import LeRobotDataset

delta_timestamps = {
    # 0.2, and 0.1 seconds *before* each frame
    "observation.images.wrist_camera": [-0.2, -0.1, 0.0]
}

# Optionally, use StreamingLeRobotDataset to avoid downloading the dataset
dataset = LeRobotDataset(
    "lerobot/svla_so101_pickplace",
    delta_timestamps=delta_timestamps
)

# Get frame in the dataset by their index
sample = dataset[0]
print(sample)
# {
# 'observation.state': tensor([...]),
# 'action': tensor([...]),
# # extra dimension due to delta timesteps
# 'observation.images.wrist_camera': tensor([3, C, H, W])
# ...
# }

batch_size=16
# wrap the dataset in a DataLoader for training/inference purposes
```

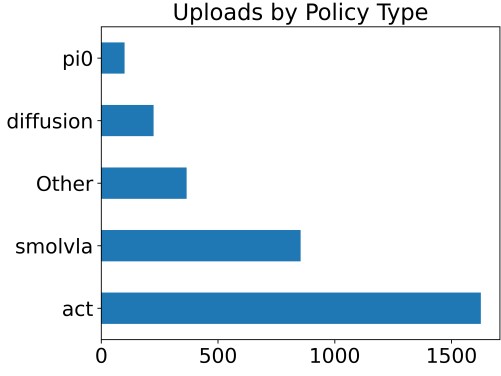 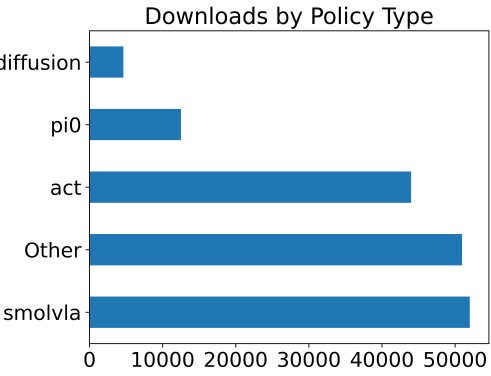

(a) Models uploaded by policy type. Policies not present have not been publicly uploaded.

(b) Models downloaded by policy type. Policies not present have not been publicly downloaded.

```python
data_loader = torch.utils.data.DataLoader(
    dataset,
    batch_size=batch_size
)

# Iterate over the DataLoader in a training loop
num_epochs = 1
device = "cuda" if torch.cuda.is_available() else "cpu"

for epoch in range(num_epochs):
    for batch in data_loader:
        # Move data to the appropriate device (e.g., GPU)
        observations = batch["observation.state"].to(device)
        actions = batch["action"].to(device)
        images = batch["observation.images.wrist_camera"].to(device)

        # Next, process the data for training or inference
        ...
```

### C.3   EXAMPLE: USE A STREAMING DATASET

```python
from lerobot.datasets.streaming_dataset import StreamingLeRobotDataset

# Streams frames on the fly without downloading the dataset
dataset = StreamingLeRobotDataset(
    "lerobot/svla_so101_pickplace",
    delta_timestamps=delta_timestamps
)
```

## D   MODELS

### D.1   EXAMPLE: TRAIN A MODEL

```python
import torch

from lerobot.configs.types import FeatureType
from lerobot.datasets.lerobot_dataset import (
    LeRobotDataset, LeRobotDatasetMetadata
)
from lerobot.datasets.utils import dataset_to_policy_features
from lerobot.policies.factory import make_pre_post_processors

# Users can use many plug-in policies from the library
from lerobot.policies.diffusion.configuration_diffusion import \
    DiffusionConfig
```

```python
13  from lerobot.policies.diffusion.modeling_diffusion import DiffusionPolicy
14
15  output_directory = "outputs/train/example_pusht_diffusion"
16  device = torch.device("cuda")
17  training_steps = 5000
18  log_freq = 1
19
20  repo_id = "lerobot/pusht" # Replace with your dataset
21  dataset_metadata = LeRobotDatasetMetadata(repo_id)
22
23  features = dataset_to_policy_features(dataset_metadata.features)
24  output_features = {
25      key: ft for key, ft in features.items()
26      if ft.type is FeatureType.ACTION
27  }
28  input_features = {
29      key: ft for key, ft in features.items()
30      if key not in output_features
31  }
32
33  cfg = DiffusionConfig(
34      input_features=input_features,
35      output_features=output_features
36  )
37
38  policy = DiffusionPolicy(cfg)
39  policy.train()
40  policy.to(device)
41  preprocessor, postprocessor = make_pre_post_processors(
42      cfg, dataset_stats=dataset_metadata.stats
43  )
44
45
46  delta_timestamps = {
47      "observation.image": [-0.1, 0.0],
48      "observation.state": [-0.1, 0.0],
49      "action": [
50          -0.1, 0.0, 0.1, 0.2, 0.3, 0.4, 0.5, 0.6,
51          0.7, 0.8, 0.9, 1.0, 1.1, 1.2, 1.3, 1.4
52      ],
53  }
54
55  dataset = LeRobotDataset(repo_id, delta_timestamps=delta_timestamps)
56
57  optimizer = torch.optim.Adam(policy.parameters(), lr=1e-4)
58  dataloader = torch.utils.data.DataLoader(
59          dataset,
60          num_workers=4,
61          batch_size=64,
62          shuffle=True,
63          pin_memory=device.type != "cpu",
64          drop_last=True,
65      )
66
67  step = 0
68  done = False
69  while not done:
70      for batch in dataloader:
71          batch = preprocessor(batch)
72          loss, _ = policy.forward(batch)
73          loss.backward()
74          optimizer.step()
75          optimizer.zero_grad()
76
77          if step % log_freq == 0:
```

```
78             print(f"step: {step} loss: {loss.item():.3f}")
79         step += 1
80         if step >= training_steps:
81             done = True
82             break
83
84 # Save a policy checkpoint.
85 policy.save_pretrained(output_directory)
86 preprocessor.save_pretrained(output_directory)
87 postprocessor.save_pretrained(output_directory)
```

### D.2 EXAMPLE: USE A PRE-TRAINED MODEL

```
1 from typing import Any
2 from lerobot.policies.smolvla.configuration_smolvla import \
3     SmolVLAConfig
4 from lerobot.policies.smolvla.modeling_smolvla import SmolVLAPolicy
5 from lerobot.datasets.lerobot_dataset import \
6     LeRobotDatasetMetadata
7
8 from lerobot.policies.factory import make_pre_post_processors
9 from lerobot.teleoperators.so100_follower.so100_follower import \
10     SO100Follower
11
12 # Take a dataset on which SmolVLA was trained, for normalization
13 repo_id = "lerobot/svla_so101_pickplace"
14 dataset_metadata = LeRobotDatasetMetadata(repo_id)
15
16 cfg = SmolVLAConfig()
17 policy = SmolVLAPolicy(cfg)
18 preprocessor, postprocessor = make_pre_post_processors(
19     cfg, dataset_stats=dataset_metadata.stats
20 )
21
22 robot = SO100Follower(...)
23 raw_obs: dict[str, Any] = robot.get_observation()
24
25 # Preprocess the observation for inference
26 policy_input = preprocessor(raw_obs)
27 # Select the action from the policy
28 policy_output = policy.select_action(policy_input)
29 # Postprocess the action for the robot
30 policy_action = postprocessor(policy_output)
31
32 robot.send_action(policy_action)
```

## E INFERENCE

Optimized inference accelerate cycle times across multiple tasks with comparable performance (Table 5), and provide a scalable path to higher model capacity without compromising on real-time control, provided access to a network. In particular, the speedup presented in Table 5 derives from *logical* decoupling—asynchronously computing the next chunk while the current one has not been exhausted yet—rather than physical decoupling, as both the server and client run on the same machine, though in principle the inference stack allows for communication between different machines.

### E.1 EXAMPLE: HOST A REMOTE SERVER

```
1 # Run this script to start the policy server on any machine
2 from lerobot.scripts.server.configs import PolicyServerConfig
3 from lerobot.scripts.server.policy_server import serve
4
5 config = PolicyServerConfig(
6     host="localhost",
```

| Inference | Success Rate (%) | | | |
|---|---|---|---|---|
| | Pick-Place | Stacking | Sorting | Avg |
| Sync | 75 | 90 | 70 | 78.3 |
| Async | 80 | 90 | 50 | 73.3 |

(a) Performance (success rates).

| Inference | Time (s) | | |
|---|---|---|---|
| | Total | Avg | Std |
| Sync | 137.5 | 13.75 | 2.42 |
| Async | 97.0 | 9.70 | 2.95 |

(b) Task completion time.

| Inference | # of Cubes | | |
|---|---|---|---|
| | Total | Avg | Std |
| Sync | 9 | 1.8 | 0.45 |
| Async | 19 | 3.8 | 1.3 |

(c) Performance in fixed time (60s per each episode).

Table 5: Comparison between regular (Sync) and optimized (Async) inference. We evaluate the SmolVLA implementation provided in `lerobot` on three real-world performed using the SO-100 arm, consisting of (1) pick and place cubes (2) stacking cubes on top of each other and (3) sorting cubes. `lerobot`'s decoupled inference schema achieves similar success rates (left) but results in significantly reduced cycle times (middle) and thus higher throughput (right), over the 10 test episodes (60s each) for the task considered.

```
7     port=8080,
8 )
9 serve(config)
```

### E.2   EXAMPLE: STREAM ACTIONS TO A ROBOT

```python
1 # Run this script to start the robot client on the robot's computer
2 import threading
3 from lerobot.scripts.server.configs import RobotClientConfig
4 from lerobot.scripts.server.robot_client import RobotClient
5
6
7 camera_cfg = ...    # cameras used by the visuomotor policy
8 robot_cfg = ...   # a given robot supported by the library
9
10 # 3. Create client configuration
11 client_cfg = RobotClientConfig(
12     robot=robot_cfg,
13     # attach to the server running the policy
14     server_address="localhost:8080",
15     # use a higher-end device for inference
16     policy_device="cuda:0",
17     policy_type="pi0",
18     pretrained_name_or_path="lerobot/pi0"
19 )
20
21 # 4. Create and start client
22 client = RobotClient(client_cfg)
23
24 task = ... # Specify the task using natural language
25
26 if client.start():
27     # Start action receiver thread
28     action_receiver_thread = threading.Thread(
29         target=client.receive_actions, daemon=True
30     )
31     action_receiver_thread.start()
32
33     try:
34         # Run the control loop
35         client.control_loop(task)
36     except KeyboardInterrupt:
37         client.stop()
38         action_receiver_thread.join()
```

