# OpenReview forum: "LeRobot:  An Open-Source Library for End-to-End Robot Learning"
_ICLR.cc/2026/Conference — ICLR 2026 Poster_

### Official Review · Reviewer_fnXV · 2025-10-30

**Soundness:** 3
**Presentation:** 1
**Contribution:** 4
**Rating:** 8
**Confidence:** 4

**Summary:**

LeRobot is an open-source, end-to-end robot-learning library from a consistent Python middleware for diverse, low-cost and humanoid/mobile robots to standardized data tooling and scalable policy deployment. It introduces the multimodal LeRobotDataset to ease large-scale collection and reuse of teleoperation data, and an optimized, decoupled inference stack that separates action prediction from execution for robust real-time control. The library ships clean PyTorch implementations of state-of-the-art methods across RL and BC (e.g., ACT, Diffusion Policy, VQ-BET, π₀, SmolVLA) with emphasize on accessibility, reproducibility, and openness. I have no doubt that it presents a useful framework for the community.

**Strengths:**

1. Many policy learning algorithms, standardized end-to-end robot learning dataset schema, tooling, and policy deployment in one framework
2. Real-robot coverage with low-cost focus for accessibility and democratization; resource transparency
3. Strong evidence of community uptake

**Weaknesses:**

The paper’s writing currently falls short of top-tier conference standards, particularly in the precision and clarity of its claims. In addition, it lacks a discussion of reproduced results from the implemented algorithms, making it difficult to assess the reliability of the framework. My detailed concerns are outlined below.

**Questions:**

1. **Scope of the “entire robotics stack.”** The claim of covering the “entire robotics stack” is an overstatement. The library primarily targets robot learning for manipulation tasks and omits major subfields such as SLAM, sim-to-real transfer, and broader perception and control systems. The authors should moderate this claim to accurately reflect the covered scope.
2. **Ambiguity in “explicit” vs. “implicit” models.** The terminology used to distinguish explicit and implicit models is confusing. The paper conflates learning-based methods with implicit models, whereas both explicit (e.g., flow models) and implicit (e.g., energy-based) formulations exist in machine learning. Clarifying what each abstraction encompasses, and specifying how methods that explicitly model the world or action distributions (with or without data) fit into this taxonomy, would improve clarity.
3. **Clarity on compounding errors.** The paper states that classical, modular methods (labeled as explicit models) suffer from compounding errors, but so do monolithic, data-driven policies (labeled as implicit models). It is unclear how the monolithic networks mitigates this issue. The authors should clarify whether their choice genuinely addresses compounding errors or simply shifts where they occur.
4. **Definition of middleware.** The role and scope of “middleware” are not clearly defined. A brief explanation in the main text of what components or functionalities are included under this term would improve clarity for readers.
5. **Language and grammar issues.** The paper would benefit from a grammar and spell check. Some (nit-pick) examples include:
    - “aboard robots” → “onboard robots” (more standard usage)
    - “parallely to low-level control” → “in parallel with low-level control loops”
    - “the the” → “the”
    - “coexistance” → “coexistence”
6. **Reproducibility of supported methods.** It is unclear whether the methods implemented in *LeRobot* reproduce the results reported in the original papers? If not, how large is the gap? Providing a clear mapping (e.g., method X on benchmark Y) and where to verify such reproduction within the repository would help assess reliability of the implemented algorithms.
7. **Extensibility and community contribution.** Including brief documentation or a section describing how to add new robots, learning methods, and simulation environments would make the framework more approachable for community contributors.
8. **Future directions.** A short discussion on planned extensions—such as supporting additional teleoperation devices, robot types, and learning methods—would help situate the library’s development trajectory. The README may be a suitable place for such a roadmap.

---

> ### Author Response · Authors · 2025-11-22
> **Comment to Reviewer fnXV (1/2)**
>
> We thank the reviewer for their precious feedback, and their acknowledgement of the relevance of providing the community with a unified end-to-end robot learning library, bridging low-level control, (standardized) dataset collection and sharing, and policy training with “clean Pytorch implementations”. We also appreciate the reviewer’s noting the library focus on low-cost real-world robots that can be independently sourced as we believe this component holds the premise of empowering the robot learning community with the same decentralized data-collection capabilities that did prove transformative in other fields of ML in the past.
>
> We sincerely thank the reviewer for their invaluable feedback regarding the presentation of our submission. As an overarching comment, we appreciate the detailed feedback provided, and have updated our submission for (1) a more precise scoping, taking in the observation that “entire robotics stack” can indeed be improved in terms of precision (2) a precise definition of middleware. We have applied all the styling changes the reviewer recommended, for which we are sincerely grateful. We thank the reviewer for their invaluable feedback on our submission, and do hope the reviewer can consider the revised version as an improvement on the points they raised.
> We here wish to address the concerns raised by the reviewer.
>
> > “The library primarily targets robot learning for manipulation tasks and omits major subfields [...]”
>
> We thank the reviewers for their precise and detailed feedback. While we feel both the title and main text of the paper specifically contextualize our library within the field of end-to-end robot learning, we appreciate how toning down our claims by referring to the “entire robot learning stack” instead of “entire robotics stack” makes our presentation more solid and correct. We immensely thank the reviewer for raising this point, giving us the opportunity to improve our submission by making it clear that the focus of the library at this stage is indeed robot learning exclusively.
>
> > “The role and scope of “middleware” are not clearly defined”
>
> We appreciate the reviewer’s feedback on the lack of a clear definition of the role and scope of  middleware. We have now updated the main body of our submission to include a precise and synthetic definition of “middleware” in the context of the library. We sincerely thank the reviewer for having raised this point regarding the clarity of our contribution.
>
> > “It is unclear whether the methods implemented in LeRobot reproduce the results reported in the original papers”
>
> We share the reviewer’s concern with ensuring reliability and reproducibility of the results, and wish to highlight that these are key areas of focus for the library.
> All the algorithms currently implemented in the library are either (1) adapted from open-source releases of the underlying methods (e.g., ACT, PI0) or (2) (co-)developed with support from the researchers and engineers proposing the method (e.g., Diffusion Policy, GR00T-1.5).
> The lack of a standard benchmark accepted by the entire robot learning community makes it challenging to verify the performance of policies and ensure reproducibility in the multitude of (real-world or simulated) benchmark embodiments and tasks used by the different methods (e.g., ACT uses the ALOHA robot setup for tasks including battery slotting, Diffusion Policy uses Franka arms to spread tomato sauce on a pizza). It is thus a core area of focus of the library to develop simulation environments the community can use for benchmark purposes (e.g., LIBERO and Meta-World), providing a unified evaluation pipeline which we hope more practitioners will adopt, resulting in clearer comparisons between methods. As per points 7 and 8, we wish to provide detailed answers to these points concerning extensibility by pointing to the library’s official documentation.
>
> ...
> (1/2)

---

> > ### Author Response · Authors · 2025-11-22
> > **Comment to Reviewer fnXV (2/2)**
> >
> > We also wish to provide an answer to some of the questions asked by the reviewer.
> >
> > - _Explicit vs Implicit models?_
> >
> > When talking about explicit and implicit models, we don’t refer to this difference in the traditional ML sense the reviewer referred to, and understand the terminology is relatively uncommon in the broader ML field, which is why we cite Bekris et al, 2024 when introducing these terms in an attempt to disambiguate. In our submission, the distinction is made between approaches which model robot-environment interactions explicitly (using, for instance, information about the robot’s kinematics) or implicitly (typically, learning-based techniques rely only on interaction data without requiring knowledge of the robot’s kinematics). This difference is crucial as it is typically considered prohibitive to describe with a high degree of fidelity robot-environment interactions in a dynamic environment where the robot performs a complex task (e.g., folding a piece of clothing, locomoting over partially irregular terrains, etc.).
> >
> > - _Compounding errors_
> >
> > We agree with the reviewer’s comments on the fact that both explicit and implicit methods do suffer from compounding errors. Rather than suggesting the opposite, in our submission we intended to make the point that designing methods based on implicit methods (i.e., robot learning contributions) may overcome these compounding errors without requiring increased modeling efforts. Zhao et al., 2023 provide evidence of how learning-based approaches can be used to overcome compounding errors by leveraging real-world interaction data only.
> >
> > We feel like thanking the reviewer for their insightful, constructive and precious feedback, which we hope to have used to improve our submission to a level where the reviewer would feel their arguments have been answered satisfactorily.

---

### Official Review · Reviewer_aYVW · 2025-11-02

**Soundness:** 4
**Presentation:** 3
**Contribution:** 4
**Rating:** 8
**Confidence:** 4

**Summary:**

This paper introduces LeRobot, an open-source library for robot learning that provides an end-to-end stack for scalable robotics research. It features unified robot integration, standardized datasets, optimized inference, and efficient, reusable state-of-the-art robot learning algorithms. This effort addresses fragmentation in the field, where researchers typically develop tools for their own use with specific robot platforms, data formats, and learning algorithms. By providing a unified and accessible framework, the library reduces the entry barrier and accelerates progress in robot learning through improved accessibility, scalability, and reproducibility.

**Strengths:**

- The paper addresses the important problem of entry barriers in robotics and embodied AI research. A unified platform and interface for hardware, data collection, and learning would benefit many researchers in the field of robot learning.

- The paper identifies key challenges and roadblocks to progress in robot learning, which motivates the proposed library.

- Current downloads and usage of the platform already demonstrate the value and need for such a unified platform. The statistics on model and dataset downloads also provide insights into the community’s interests and needs over time.

**Weaknesses:**

- Lack of discussion of what tasks or scenarios may not be suitable for the current LeRobot library. Would all researchers in robot learning benefit from the current LeRobot library?
- The paper would benefit from more discussion on the integration of simulated environments, as many researchers start with toy simulations. What is the procedure for adding custom environments to the ecosystem?
- Broken citation on line 266.

**Questions:**

- How does LeRobot compare to ROS? What factors might make LeRobot more successful than ROS as an open-source robotics platform?
- How is the correctness of the implemented algorithms ensured, and how is reproducibility maintained?

---

> ### Author Response · Authors · 2025-11-22
> **Comment to Reviewer aYVW (1/2)**
>
> We are delighted by the reviewer's acknowledgement of the relevance of providing a unified, unfragmented and open-source set of tools for the field of robot learning. We are especially pleased by the reviewer’s acknowledgement of the library’s potential in accelerating robot learning research. Providing a valuable tool to the research community is indeed a core objective of the library. We share the reviewer’s judgement relative to how usage metrics (Figures 5 and 7) seem to indicate the community’s interest in a unified and accessible framework for robot learning in general, together with the increased level of interest in generalist control policies adapted for different more specific tasks (Figure 7(d)). We hope to address the reviewer’s concerns below.
>
> > “Lack of discussion of what tasks or scenarios may [...]”
>
> We appreciate the reviewer’s candor in sharing that they think we have not communicated with enough detail the shortcomings of the current status of the library. While we have explicitly listed limitations of the current status of the library in §4, we appreciate the role that a more in detail discussion of the limitations of lerobot can have to guide further development of the library. We wish to point to the project official release page where we regularly engage with the open-source community in discussing implementation as well as conceptual shortcomings of the library. In general, we believe the positioning of the library to be ideal to contribute to the large majority of robot learning research efforts, and think the large majority of researchers working in end-to-end robot learning as well as on specific sub-components in a usual robotics pipeline (perception, control, etc.) can benefit from it.
>
> > “more discussion on the integration of simulated environments”
>
> We agree with the reviewer’s take regarding the lack of discussion of simulation frameworks, which we have not included in the main body of our submission as (1) the main focus of the library is real-world robotics and (2) simulation environments (LIBERO and MetaWorld) are mainly used for evaluation purposes. We have now updated our submission with information on simulation environments in Appendix §E, and hope to be able to move this section to the main body of the text should this submission be accepted and thus the page limit extended from 9 pages to 10. With that being said, we still wish to highlight (1) the library’s core focus on real-world robotics over simulation environments, which we support for accessibility while noting that we believe easily-sourceable robots (e.g., fully 3d printable such as the SO-10X) provide an entry-point to real-world robotics without the fundamental challenges of crossing the reality gap (e.g., [[1]](https://arxiv.org/abs/1703.06907), [[2]](https://arxiv.org/abs/1910.07113)). Regarding the procedure to add custom environments, we wish to provide a detailed guide of how to include more environments by pointing to the library’s official documentation.
>
> ...
> (1/2)

---

> > ### Author Response · Authors · 2025-11-22
> > **Comment to Reviewer aYVW (2/2)**
> >
> > ...
> > We also wish to answer the reviewer’s specific questions.
> > - _LeRobot vs. ROS_
> >
> > We could not answer this question without first noting the very sizable impact the ROS open-source community has had on robotics, and by noting how we view LeRobot and ROS as complementary efforts to accelerate robotics rather than opposite projects. With that being said, LeRobot is indeed more biased towards robot learning-based techniques than ROS2, which, under the assumption that ML, can indeed prove transformative in robotics, could be a key factor in LeRobot’s potential success. In terms of the open-source nature of the projects, LeRobot is notably written in Python, and thus benefits from the large community of researchers and engineers already proficient with the language.
> >
> > - _How is reproducibility maintained?_
> >
> > All algorithms currently implemented in the library are either (1) adapted from open-source
> > releases of the underlying method to ensure compatibility with the library’s API (e.g., ACT, PI0), (2) (co-)developed with support from the researchers and engineers proposing the method (e.g., Diffusion Policy, GR00T-1.5). As far reproducibility goes, we wish to observe how the lack of a standard benchmark accepted by the robot learning community makes it challenging to verify the performance of policies against the multitude of benchmark tasks and embodiments used by different methods (e.g., ACT uses the ALOHA setup and tasks such as battery slotting, while Diffusion Policy uses a Franka arm and tasks such as spreading tomato sauce on a pizza). For this precise reason, it is a core area of focus of the library to develop simulation environments the community can use for benchmark purposes (e.g., LIBERO and Meta-World) to ensure higher reproducibility across implementations of the same method. Our submission has now been modified to specifically include simulation environments (Appendix §E) so that users of the library can aim for a higher degree of reproducibility across implementations. We thank the reviewer for providing valuable input on this rather crucial matter for the field of robot learning.
> >
> > We are extremely grateful for the reviewer’s detailed feedback, and remain very available to address any standing concerns the reviewer may have.
> >
> > (2/2)

---

### Official Review · Reviewer_5MwV · 2025-11-03

**Soundness:** 3
**Presentation:** 3
**Contribution:** 4
**Rating:** 6
**Confidence:** 5

**Summary:**

The paper presents a library for end-to-end robot learning. It addresses a very pervasive and relevant problem of a typically fragmented robotics stack, for which having simple and unified tooling would greatly enhance productivity and rampup time for beginners. It has various subcomponents that address different parts of the stack: hardware integration, local and streaming datasets in a shared format, async inference for action chunking based models, RL and IL constructs, simulation integrations etc. All of these together are an excellent starting point for someone looking to conduct research and development on low-cost readily available robots in a non-production setting.

On balance, I am slightly leaning towards accepting the paper as it would provide a reference to people using the tooling to conduct research and for publications. However, I do feel that the core functionality of the library has several avenues for improvement (some iterative and others fundamental).

**Strengths:**

- This is clearly a significant contribution that will help a lot of additional people easily take up robotics and experiment on well-integrated low cost robots. Someone who would previously have been blocked on a part of a complicated stack can use the abstracted tooling of the library to get started quicker and with lesser frustration.

- The library supports a lot of popular models and algorithms that have been shown to work on real robots in a variety of settings.

- The library contains support for asynchronous inference (i.e. that used in modern models that use action chunking and related methods). The functionality to use a hybrid cloud-robot inference is also very useful.

- The dataset tooling seems very useful, especially the structure which helps handle larger scale datasets with millions of trajectories.

- I find it positive that the paper explicitly lists the limitations of the library. This is absolutely understandable since it is in active development.

**Weaknesses:**

- While the overall library seems excellent as a starting point for someone learning about robotics or who wants to get started with minimal friction with a toy problem, it seems pretty raw for something like production usage. The parallel I would like to draw here is of pytorch, which was suited for both simplicity and production robustness.

- Local porting of large datasets seems very slow compared to what should be possible. E.g. 7+ days of processing time for DROID should be able to be sped up significantly by better parallel processing and systems engineering.

- I feel like LeRobot overindexes on what is ‘popular’ in the current robotics/ML types of models and algorithms (e.g. RL and imitation learning), but lacks some essential functionality that a user would likely need to resort to a third party library for. An example of this is optimized motion planners or IK solvers that could run in either real or simulation environments, which could give ground truth data to train particular e2e models.

- Simulation is often a critical component of end-to-end robot learning. I would have liked to see additional support for additional simulation frameworks (e.g. IsaacSim or Genesis). This to me, would abstract away a lot of the friction a typical user needs to undergo to setup the simulation environment itself. Note: MetaWorld and Libero seem impractical for a lot of real world tasks compared to something more comprehensive.

- A crucial element of deploying robotic systems in the real world is a robust safety layer. In my opinion, the deployment portion of the library should treat safety as the most important principle and contain abstractions that allow for this to be realized (e.g. add protective stops based on certain criteria). While these are typically also implemented in the firmware of the robot itself (e.g. as protective stops or estops), having an added layer on top that can focus on the specific logical safety considerations would be helpful when directly coupled with (for example) the RobotClient / PolicyServer

- (not relevant to the assessment of the paper) It was impossible to not know where the work came from, given that the library is already very popular in the online robotics/ML community.

**Questions:**

- Do you have any detailed benchmarks for the throughput of LeRobotDataset and StreamingLeRobotDataset as opposed to something like MCAP?

- Do you plan to add GELLO teleop support?

---

> ### Author Response · Authors · 2025-11-22
> **Comment to Reviewer 5MwV (1/2)**
>
> We are delighted the reviewer acknowledges the library’s impact in (1) providing a unified set of toolings for robot learning and (2) powering research efforts by allowing fast prototyping. Our intention is indeed to develop a solid suite of tools researchers and practitioners can use, and we are happy the reviewer recognizes our efforts in curating (1) a scalable dataset format (2) implementations of multiple SOTA algorithms and (3) software tools for better use in practice. We hope to address the reviewer’s concerns below.
>
> We sincerely appreciate the reviewer’s critical feedback, whose value is further underscored by the fact similar points have been raised recently, and have thus become ongoing efforts of both the library’s core maintainers and open-source contributors. We here wish to address the weaknesses mentioned by the reviewer, which we found very valuable and agreeable. Nonetheless, we still believe that even with these limitations our work does represent a contribution toward a unified end-to-end stack for robot learning.
>
> > “Library is pretty raw for production usage”
>
> The library’s usage in production-ready use cases is indeed rather limited at this stage, in keeping with our intention to develop it as a tool for researchers and early adopters first. That said, we believe both efforts presented in the main text of the paper (e.g., scalable dataset architecture §3.2, hybrid cloud-local support §3.4) and in the Appendix (e.g., ease of use of pre-trained models, §D.2) can be considered significant steps in the direction of developing a production-ready library, as both training on large-scale datasets and using large-scale models under computational constraints can have a significant impact in more production-oriented use cases. In terms of software architecture, we already support easy-to-use, simple CLI instructions to perform training (lerobot-train …), evaluation (lerobot-eval …), recording of a dataset (lerobot-record …) and more, and hope to extend and improve the coverage of these instructions targeting production use-cases over research.
>
> > “Overindexing on popular methods within robotics, and lack of coverage of traditional methods”
>
> We agree with the reviewer on the library’s bias towards robot learning techniques over more traditional methods. While very aware of the relevance of more traditional approaches to robotics, it is our belief that learning-based approaches to robotics hold the premise of much greater scalability, particularly in the context of the growing availability of (open) robotics data. We believe the focus of the library to have been opportunely communicated in our submission, and remain more than willing to further modify our presentation to make it even clearer that the library’s main focus is indeed robot learning.
>
> > “Lack of support for simulation frameworks”
>
> We agree with the reviewer’s take on the relevance of simulation frameworks for robot learning, and did not explicitly include our contributions in terms of simulations (LIBERO and MetaWorld) for reasons of space. We wish to highlight (1) the library’s core focus on real-world robotics over simulation environments, which we do support although they are not our main area of focus and that (2) efforts made by both the core maintainers of the library and the broader open-source community to extend support to more simulation frameworks (Genesis, IsaacSim, etc) are currently ongoing. We have now updated our submission with information on simulation environments in Appendix §E, and hope to be able to move this section to the main body of the text should this submission be accepted and thus the page limit extended to 10 pages.
>
> > What is the safety layer for deployment”
>
> We share the reviewer’s concern for safety, and wish to use this opportunity to highlight how operational safety is of the utmost priority for the library. Besides being implemented at the firmware level, the library implements high level checks to prevent unsafe commands from being executed. For instance, (1) the library implements limits on the maximal torque that can be applied on some joints of some robot platforms (e.g., 50% maximal torque on the gripper of SO-10X in order to prevent motor burnout) and (2) prevents the execution of certain commands when SO-10X platforms are powered at 12V. Our interest in developing the library in the open is to also provide the community with a set of open tools to contribute to safety research in the context of robot learning. We hope to achieve this goal by providing a platform for weaknesses to emerge, be studied and addressed in the open.
>
> ...
> (1/2)

---

> > ### Author Response · Authors · 2025-11-22
> > **Comment to Reviewer 5MwV (2/2)**
> >
> > …
> > > “Porting of large scale datasets is slow and non optimized”
> >
> > We share the reviewer’s interest in optimizing the performance of components of the library such as porting datasets to the format defined in the library. We wish to highlight the heterogeneity of robotics data formats (TFDS, ROS bags, JSON, etc.) represents a significant obstacle towards more optimized and less bespoke solutions for porting large amounts of data from any format to the library’s. Nonetheless, the library does already support running conversion jobs in a distributed fashion on clusters, which can significantly reduce conversion time. We have also begun investigating more optimized porting mechanisms (e.g., aggregating multiple datasets in parallel rather than sequentially). Alongside other optimizations, the current status of the library allows it to port a large dataset such as Behavior1K (~2TB) in ~2 hours using ~32 concurrent CPU-only jobs on.
> >
> > We also wish to address the reviewer’s specific questions:
> >
> > - _(Streaming)LeRobotDataset versus MCAP_
> >
> > We do not currently have a benchmark for other low-level data formats, and in general we limit our focus to the format natively supported by `datasets`, a widely adopted and popular (21k+ stars on github) library to openly share datasets. We operate under this constraint to lower the barrier of entry as much as possible. Despite this, we wish to highlight that we trust the community to extend support of (Streaming)LeRobotDatasets to different modalities whenever the performance downsides related to using datasets should outweigh the benefits of being able to easily share possibly very diverse datasets via a common structure. In this, we are explicitly prioritizing the decentralized nature of the project over the performance of the individual components.
> >
> > - _GELLO teleop support_
> >
> > We wish to take this opportunity to recognize the relevance of GELLO for the field to enable affordable teleoperation. As we currently do not support any of the higher-end robots for which GELLO was first designed (Franka, UR5, xArm), we are not planning to extend support to GELLO in the near future, or at least as long as cheap, affordable platforms such as the SO-10X platforms remain our focus. However, due to GELLO’s significance in the context of teleoperation, we now (duly) added it to the citations in the body of the paper, and thank the reviewer for their precious input on this matter.
> >
> > We are extremely grateful for the reviewer’s detailed and constructive feedback, and feel it drastically improved the quality of our submission. We remain very available to address any standing concerns the reviewer may have.
> >
> > (2/2)

---

### Meta-Review · Area_Chair_2Wc8 · 2025-12-30

**Summary:**

Reviewers generally agree that LeRobot is a valuable open-source infrastructure contribution that addresses fragmentation in robot learning by integrating hardware interfaces, dataset tooling, learning algorithms, and deployment abstractions into a unified, accessible framework. The main concerns influencing the decision relate to scope and positioning (e.g., overstatements about covering the “entire robotics stack”), presentation clarity and terminology, limited explicit evidence for reproducibility and correctness of implemented algorithms, and the current breadth and maturity of ecosystem support (simulation frameworks, traditional robotics components, and production-level robustness). Overall, reviewer sentiment is positive, with concerns focused mainly on clarity, documentation, and scope rather than the core contribution.

**Reviewer Concerns:**

The rebuttal addresses several key concerns by clarifying the intended scope of the library (robot learning rather than the full robotics stack), refining definitions and terminology, improving the presentation, expanding the discussion of simulation support, and explaining design choices regarding the learning-based focus, safety checks, and dataset conversion performance. However, some concerns remain partially outstanding, particularly the lack of concrete, systematic evidence in the paper for reproducing results or verifying the correctness of the implemented algorithms, as well as the limited current breadth of simulation support and absence of traditional robotics components, which the authors frame as intentional scope choices. Concerns about production readiness remain, but are less critical given the stated research-focused target audience.

**Reviewer Scores:**

Based on the rebuttal, Reviewer 5MwV would likely remain at 6, as the response constructively addresses concerns but leaves several scope and maturity issues unresolved. Reviewer aYVW would likely remain at 8, with the rebuttal reinforcing an already positive assessment. Reviewer fnXV would likely also remain at 8, with improved confidence due to clarified scope and presentation, but with remaining reservations about explicit reproducibility evidence.

---

### Decision · Program_Chairs · 2026-01-26

Accept (Poster)